# Diabetic retinopathy as a predictor for peripheral compression neuropathies, a registry-based study

Caroline Olsson[1,2], Mattias Rydberg[2,3], Malin Zimmerman[1,2]*

**1** Department of Orthopedics, Helsingborg Hospital, Helsingborg, Sweden, **2** Department of Translational Medicine–Hand Surgery, Lund University, Malmö, Sweden, **3** Department of Hand Surgery, Skåne University Hospital, Malmö, Sweden

* malin.zimmerman@med.lu.se

## Abstract

Diabetes is characterized by hyperglycaemia and entails many complications, including retinopathy and entrapment neuropathies, such as ulnar nerve entrapment (UNE) and carpal tunnel syndrome (CTS). Hyperglycaemia damages the nerves of the retina, as well as peripheral nerves. There is a correlation between entrapment neuropathies and retinopathy in patients with diabetes, but whether patients with diabetic retinopathy are more prone to develop CTS and UNE is uncertain. Hence, the aim was to investigate if retinopathy can be used as a factor predicting the development of CTS and UNE. Data from 95,437 individuals from the National Diabetes Registry were merged with data from the Skåne Healthcare Registry. The population was analysed regarding prevalence of CTS or UNE and retinopathy status. Population characteristics were analysed using the Chi²-test, Student's Independent T-test, and the Mann-Whitney U-test. Two logistic regression models were used to analyse the odds ratio (OR) for development of CTS and UNE depending on retinopathy status, adjusted for possible confounders. Both CTS and UNE were more frequent among those with retinopathy, compared to those without (CTS: 697/10,678 (6.5%) vs. 2756/83,151 (3.3%; p<0.001), (UNE: 131/10,678 (1.2%) vs. 579/83,151 (0.7%; p<0.001)). The OR for developing CTS for individuals with type 1 diabetes and retinopathy was 2.40 (95% CI 2.06–2.81; p<0.001) and of developing UNE was 1.53 (0.96–2.43; p = 0.08). The OR for developing CTS for individuals with type 2 diabetes and retinopathy was 0.93 (0.81–1.08; p = 0.34) and for UNE 1.02 (0.74–1.40; p = 0.90). Diabetic retinopathy is associated with a higher risk of developing CTS and UNE, but the association seems to be mediated by the duration of the diabetes. Higher HbA1c levels, longer diabetes duration and higher BMI are significant risk factors for developing CTS and UNE in type 1 and type 2 diabetes.

## Introduction

Both type 1 diabetes (DM1) and type 2 diabetes (DM2) are associated with several complications. Over time, cardiovascular disease, nephropathy, neuropathy, and retinopathy may develop [1]. Retinopathy is a microvascular complication affecting the eye, and has a

**Data Availability Statement:** Public access to data is restricted by the Swedish Authorities (Public Access to Information and Secrecy Act; https://government.se/information-material/2009/09/public-access-toinformation-andsecrecy-act/) but data can be made available to researchers after a

special review that includes approval of the research project by both the National Ethics Committee (application through https://etikprovningsmyndigheten.se) and the authorities' data safety committees (https://registercentrum.se/vara-tjaenster/datauttag/p/B1eeV66QE and https://rcsyd.se).

**Funding:** This work was supported by grants from Region Skåne (MR), Lund University (CO, MZ), Kockska stiftelsen (MZ), Stig and Ragna Gorthon Foundation (MR and MZ), Swedish Diabetes Foundation [DIA2020-492] (research group), Swedish Research Council (research group), Regional Agreement on Medical Training and Clinical Research (ALF) between Region Skåne and Lund University (MZ) and Magnus Bergvall Foundation [2020-03612] (MZ). The funders had no role in study design, data collection and analysis, decision to publish, or preparation of the manuscript.

**Competing interests:** The authors have declared that no competing interests exist.

prevalence of about 40% in people with diabetes [2]. Neuropathies, including entrapment neuropathies (EN), are also common in individuals with diabetes, the most frequently occurring being carpal tunnel syndrome (CTS), which affects 10–14% of people with diabetes, and ulnar nerve entrapment at the elbow (UNE), which affects 2% of people with diabetes [3].

The microvascular damage in diabetes results in a restricted blood flow to nerves which are sensitive to ischemia [4, 5]. This may contribute to development of neuropathies such as retinopathy and EN in people with diabetes [5].

The risk of developing both retinopathy and neuropathies increases with higher levels of HbA1c over time, and can be reduced by lowering HbA1c [4, 6].

As both retinopathy and entrapment neuropathies are consequences of microvascular damage and have a somewhat similar pathology, it is possible that signs of retinopathy could be used as a predictor for the risk of developing entrapment neuropathies such as CTS and UNE.

Hence, the primary aim of this study was to investigate if retinopathy is a predisposing factor for the development of CTS and UNE in people with DM1 and DM2. The secondary aim was to investigate the impact of other potential risk factors related to diabetes for developing EN including BMI, duration of diabetes, levels of HbA1c and blood lipids.

## Methods

### Study design

This is a retrospective cohort study based on data from the Skåne Healthcare Register (SHR) and the Swedish Diabetes Register (Nationella Diabetesregistret; NDR) concerning people living in Southern Sweden during the study period. Data from patients in the SHR containing information about diagnosis of CTS or UNE between 2004 and 2019 were merged with data from the NDR on patients with diabetes mellitus.

### NDR

The NDR was created in 1996 and contains data on clinical characteristics and prevalence of complications, from patients over 18 years of age with diabetes. The NDR is a national quality register and covers about 90% of all patients with diabetes in Sweden [7, 8]. The data are reported to the NDR continuously by both primary care and specialized clinics [9]. We included patients with a diagnosis of DM1 or DM2. Patients with other types of diabetes, such as gestational diabetes and diabetes secondary to pancreatitis, were excluded. The NDR provided data about type and duration of diabetes, sex, age, retinopathy (based on fundus photography), HbA1c, whether the patient was a smoker or not, BMI, systolic blood pressure, glomerular filtration rate (GFR), creatinine and levels of LDL, cholesterol and triglycerides.

### SHR

The SHR is a regional register containing information about all individual care contacts from patients in the region of Skåne in Southern Sweden [10]. The data are registered and entered into the register by primary care clinics, specialized clinics and inpatient care, in both the private and public sectors. The register contains information about when and where the care was given, the patient's medical diagnoses, and general information about the patient such as age and sex. Patients were identified in the SHR using ICD-10-codes G562 (UNE) and G560 (CTS).

### Statistics

We excluded individuals with prevalent CTS or UNE, i.e., a diagnosis of CTS or UNE previous to their diagnosis of DM. Included patients were divided into three groups based on HbA1c

levels: <48mmol/mol, 48-64mmol/mol and >64mmol/mol, according to an adaption of the Swedish, American and NICE guidelines regarding desirable levels of HbA1c [11–14]. We considered that an individual had retinopathy if the diagnosis of retinopathy was present before the diagnosis of CTS or UNE. Smoking and age were recorded at baseline. BMI, systolic blood pressure, HbA1c as a continuous variable, GFR, creatinine and blood lipids levels are presented as means of the cumulative values over the study period.

Nominal data are presented in figures (%). Normally distributed data are presented as mean ± standard deviation (SD). Non-normally distributed data are presented as a median [quartile 1; Q1- quartile 3; Q3].

Categorical, nominal data were analysed using crosstabs for number and percent of patients, and the $Chi^2$-test was used for significance testing between groups. Quantitative data were analysed using histograms, mean, median and quartiles to determine distribution. Normally distributed data were analysed using Student's Independent T-test for significance testing and non-normally distributed data were analysed with the non-parametric Mann-Whitney U-test.

Binary logistic regression models were used to study the effect of retinopathy on the risk of developing CTS or UNE, stratified for DM1 and DM2. In the first model, retinopathy status, sex and age were included. In the second model, HbA1c-groups, smoking, BMI, levels of LDL, systolic blood pressure and duration of diabetes were added. The variables used in the logistic regressions were all based on previously known risk factors for the development of EN, such as elevated HbA1c and duration of diabetes, and variables included in the metabolic syndrome (MetS) [4, 5, 15–18]. LDL was used instead of triglycerides and cholesterol, as LDL is the harmful type of cholesterol and is the most likely of the three to be associated with EN, according to previous studies [16]. Pearson correlation was used to investigate correlations between variables.

A p-value <0.05 was considered statistically significant. IBM SPSS statistics for Mac, version 27 (SPSS inc., Chicago, Illinois, USA) was used for all calculations.

## Ethics

This study was approved by the Regional Ethical Review Board in Lund, Sweden (2019–02042). Patients provide written informed consent before they are included in the NDR.

## Results

We identified 99,901 individuals in the SHR and NDR. We excluded 4,464 individuals due to secondary or unidentifiable diabetes type or missing data on diabetes type. We then excluded 1,608 individuals with a diagnosis of CTS/UNE prior to their diagnosis of DM, resulting in a study population of 93,829 individuals. A flowchart of the inclusion process is presented in Fig 1.

The study population comprises 10,678 (11%) individuals with retinopathy in 3069/9645 (32%) individuals with type 1 diabetes and 7609/84,184 (9%) individuals with type 2 diabetes.

Both CTS and UNE were more common among individuals with retinopathy than among those without retinopathy (CTS: 697/10,678 (6.5%) vs. 2,756/83,151 (3.3%; p<0.001), (UNE: 131/10,678 (1.2%) vs. 579/83,151 (0.7%; p<0.001)).

## Type 1 diabetes

The study population included 9645 individuals with DM1. The group of individuals with retinopathy and DM1 were older, had a longer diabetes duration, higher HbA1c levels, higher systolic blood pressure, worse kidney function and a significantly higher incidence of CTS and UNE than individuals with DM1 but without retinopathy (Table 1). Retinopathy, female sex, higher HbA1c,

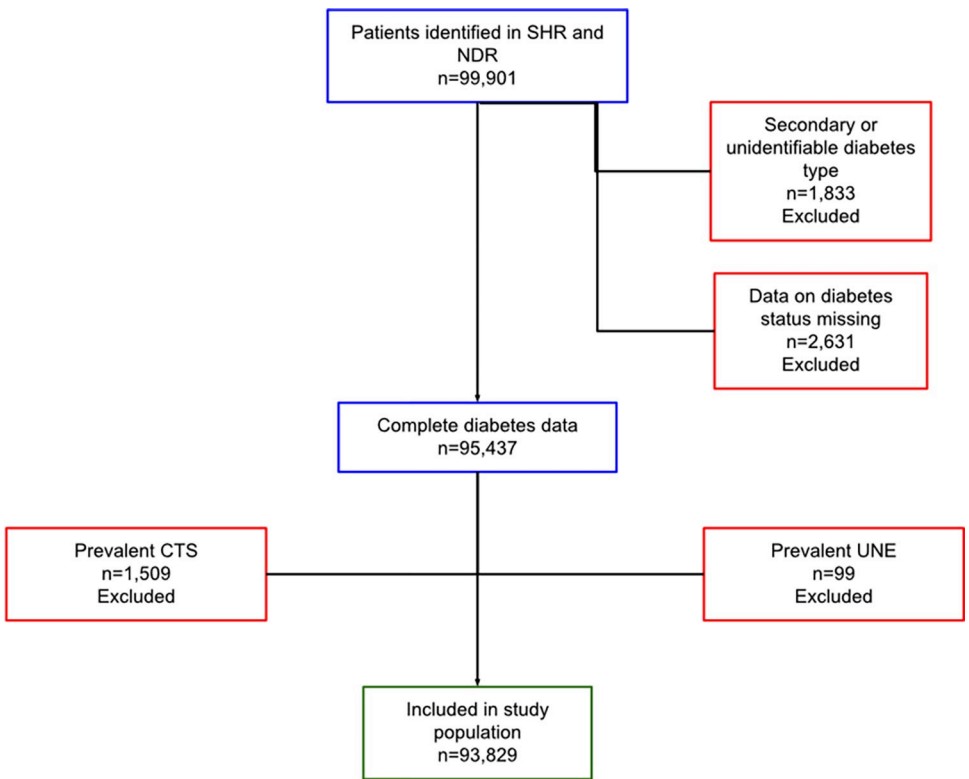

**Fig 1. Flowchart describing the inclusion process.** CTS = carpal tunnel syndrome, UNE = ulnar nerve entrapment at the elbow.

**Table 1. Clinical characteristics in a population with diabetes type 1 (n = 9645) evaluated for carpal tunnel syndrome (CTS) and ulnar nerve entrapment at the elbow (UNE).**

| | Retinopathy n = 3069 | No retinopathy n = 6576 | P-value |
|---|---|---|---|
| **Women** | 1368 (45) | 2915 (44) | 0.82 |
| **Age at baseline, years** | 58±18 | 49±20 | <**0.001** |
| **Duration of diabetes, years** | 37 [27–47] | 17 [9–25] | <**0.001** |
| **Smoking** | 522 (18) | 873 (17) | 0.056 |
| **HbA1c, mmol/mol** | 65±11 | 62±13 | <**0.001** |
| **BMI, kg/m²** | 25.9±4.1 | 25.7±4.5 | 0.22 |
| **Systolic BP, mmHg** | 131±12 | 127±13 | <**0.001** |
| **Triglycerides, mmol/l** | 1.0 [0.7–1.3] | 0.9 [0.7–1.3] | 0.088 |
| **Total LDL, mmol/l** | 2.4±0.7 | 2.4±0.7 | **0.045** |
| **Total cholesterol, mmol/l** | 4.5±0.8 | 4.5±0.8 | 0.38 |
| **Creatinine, μmol/l** | 76 [65–92] | 72 [62–82] | <**0.001** |
| **GFR, ml/min/1.73m²** | 83±28 | 96±25 | <**0.001** |
| **Incident CTS** | 377 (12.3) | 346 (5.3) | <**0.001** |
| **Incident UNE** | 67 (2.2) | 57 (0.9) | <**0.001** |

Values presented as a number (%), mean±standard deviation if normally distributed and median [interquartile range] if non-normally distributed. Data on smoking status missing in 1572 individuals. BMI; body mass index, BP; blood pressure, GFR; glomerular filtration rate, LDL; low density lipoprotein.

longer diabetes duration and higher BMI were all strong predictors for developing CTS in the logistic regression model (Table 2). Diabetes duration and higher BMI were significant predictors for developing UNE in the logistic regression model (Table 3). When studying correlations between different factors, we found weak correlations between CTS and UNE, and between retinopathy and CTS as well as between duration of diabetes and CTS (Table 4).

## Type 2 diabetes

There were 84,184 individuals with DM2. There were more men and more smokers in the group with retinopathy and DM2 than in the group without retinopathy and DM2 (Table 5). The group with retinopathy and DM2 were also older, had longer diabetes duration, higher HbA1c levels, higher systolic blood pressure, worse kidney function and higher incidence of CTS than the group without retinopathy (Table 5). The group of individuals with retinopathy and DM2 had slightly better blood lipid levels and lower BMI than the group without retinopathy (Table 5). In the logistic regression model, female sex, higher HbA1c levels, longer diabetes duration and higher BMI were significant predictors for CTS (Table 6). HbA1c>64mmol/mol, smoking, longer diabetes duration and higher BMI were predictors for UNE in individuals with DM2 (Table 7). In the Pearson correlations, we found a weak correlation between CTS and UNE in individuals with DM2 (Table 8).

## Discussion

In this retrospective register-based cohort study, CTS and UNE occurred more commonly among individuals with diabetic retinopathy than among individuals without. In the

**Table 2. Logistic regression model predicting the odds for developing carpal tunnel syndrome in people with diabetes type 1 (n = 9645).**

|  | Odds Ratio (95% CI) | P-value |
|---|---|---|
| *Model 1* | | |
| **Age** | 1.01 (1.00–1.01) | **<0.001** |
| **Male** | *Reference* | |
| **Female** | 2.19 (1.87–2.56) | **<0.001** |
| **No retinopathy** | *Reference* | |
| **Retinopathy** | 2.40 (2.06–2.81) | **<0.001** |
| *Model 2* | | |
| **Age** | 0.98 (0.99–1.00) | 0.32 |
| **Male** | *Reference* | |
| **Female** | 2.08 (1.74–2.48) | **<0.001** |
| **No retinopathy** | *Reference* | |
| **Retinopathy** | 1.63 (1.33–1.99) | **<0.001** |
| **HbA1c <48mmol/mol** | *Reference* | |
| **HbA1c 48–64 mmol/mol** | 1.76 (1.13–2.73) | **0.013** |
| **HbA1c >64 mmol/mol** | 2.02 (1.30–3.15) | **0.002** |
| **Non-smoker** | *Reference* | |
| **Smoker** | 1.04 (0.82–1.30) | 0.77 |
| **Diabetes duration** | 1.03 (1.02–1.04) | **<0.001** |
| **Systolic blood pressure** | 1.00 (0.99–1.00) | 0.37 |
| **BMI** | 1.06 (1.04–1.08) | **<0.001** |
| **LDL** | 0.98 (0.86–1.12) | 0.81 |

BMI; body mass index, LDL; low density lipoprotein.

**Table 3. Logistic regression model predicting the odds for developing ulnar nerve entrapment at the elbow in people with diabetes type 1 (n = 9645).**

|  | Odds Ratio (95% CI) | P-value |
|---|---|---|
| *Model 1* | | |
| **Age** | 1.01 (1.00–1.02) | 0.065 |
| **Male** | Reference | |
| **Female** | 0.99 (0.69–1.42) | 0.96 |
| **No retinopathy** | Reference | |
| **Retinopathy** | 2.36 (1.64–3.40) | **<0.001** |
| *Model 2* | | |
| **Age** | 1.00 (0.99–1.01) | 0.80 |
| **Male** | Reference | |
| **Female** | 0.86 (0.58–1.29) | 0.48 |
| **No retinopathy** | Reference | |
| **Retinopathy** | 1.53 (0.96–2.43) | 0.08 |
| **HbA1c <48mmol/mol** | Reference | |
| **HbA1c 48–64 mmol/mol** | 1.99 (0.61–6.50) | 0.22 |
| **HbA1c >64 mmol/mol** | 2.63 (0.81–8.53) | 0.11 |
| **Non-smoker** | Reference | |
| **Smoker** | 1.46 (0.90–2.35) | 0.13 |
| **Diabetes duration** | 1.03 (1.01–1.05) | **<0.001** |
| **Systolic blood pressure** | 1.00 (0.98–1.02) | 0.97 |
| **BMI** | 1.05 (1.01–1.11) | **0.02** |
| **LDL** | 1.12 (0.84–1.52) | 0.44 |

BMI; body mass index, LDL; low density lipoprotein.

regression model, retinopathy was a strong predictor for CTS in DM1, but not for DM2 in adjusted models. Retinopathy did not predict the development of UNE. HbA1c, BMI and diabetes duration were significant predictors for both CTS and UNE in both DM1 and DM2. Blood lipid levels did not seem to affect incidence of CTS and UNE.

**Table 4. Correlation matrix.** Diabetes-related factors and carpal tunnel syndrome and ulnar nerve entrapment at the elbow in people with diabetes type 1 (n = 9645).

|  | CTS | UNE | Age | Retino-pathy | HbA1c | Smoking | Duration of diabetes | Systolic blood pressure | BMI | LDL | Creatinine |
|---|---|---|---|---|---|---|---|---|---|---|---|
| **CTS** | 1 | | | | | | | | | | |
| **UNE** | 0.195 | 1 | | | | | | | | | |
| **Age** | 0.061 | 0.03 | 1 | | | | | | | | |
| **Retino-pathy** | 0.124 | 0.054 | 0.222 | 1 | | | | | | | |
| **HbA1c** | 0.04 | 0.028 | -0.015 | 0.116 | 1 | | | | | | |
| **Smoking** | -0.001 | 0.017 | -0.011 | 0.021 | 0.179 | 1 | | | | | |
| **Duration of diabetes** | 0.156 | 0.071 | 0.504 | 0.534 | 0.100 | -0.033 | 1 | | | | |
| **Systolic BP** | 0.031 | 0.024 | 0.566 | 0.177 | 0.023 | 0.009 | 0.324 | 1 | | | |
| **BMI** | 0.07 | 0.027 | 0.093 | 0.013 | 0.025 | -0.073 | -0.011 | 0.195 | 1 | | |
| **LDL** | 0.003 | 0.011 | -0.033 | -0.021 | 0.111 | 0.043 | -0.065 | 0.025 | 0.097 | 1 | |
| **Creati-nine** | 0.018 | 0.014 | 0.192 | 0.187 | 0.001 | -0.027 | 0.243 | 0.20 | 0.029 | -0.073 | 1 |

Pearson correlations. BMI; body mass index, BP; blood pressure, CTS; carpal tunnel syndrome, LDL; low density lipoprotein, UNE; ulnar nerve entrapment at the elbow.

**Table 5. Clinical characteristics in a population with diabetes type 2 (n = 84,184) evaluated for carpal tunnel syndrome (CTS) and ulnar nerve entrapment at the elbow (UNE).**

| | Retinopathy n = 7609 | No retinopathy n = 76,575 | P-value |
|---|---|---|---|
| Women | 2947 (39) | 32,395 (42) | <0.001 |
| Age at baseline, years | 76±13 | 72±14 | <0.001 |
| Duration of diabetes, years | 20 [13–27] | 11 [6–17] | <0.001 |
| Smoking | 1022 (15) | 9849 (17) | <0.001 |
| HbA1c, mmol/mol | 58±13 | 53±12 | <0.001 |
| BMI, kg/m$^2$ | 29.9±5.4 | 30.1±5.4 | 0.03 |
| Systolic BP, mmHg | 138±13 | 136±12 | <0.001 |
| Triglycerides, mmol/l | 1.6 [1.2–2.2] | 1.6 [1.2–2.2] | <0.001 |
| Total LDL, mmol/l | 2.4±0.8 | 2.5±0.8 | <0.001 |
| Total cholesterol, mmol/l | 4.4±0.9 | 4.5±1.0 | <0.001 |
| Creatinine, μmol/l | 85 [70–106] | 78 [66–93] | <0.001 |
| GFR, ml/min/1.73m$^2$ | 72±26 | 79±24 | <0.001 |
| Incident CTS | 320 (4.2) | 2410 (3.1) | <0.001 |
| Incident UNE | 64 (0.8) | 522 (0.7) | 0.11 |

Values presented as numbers (%), mean±standard deviation if normally distributed and median [interquartile range] if non-normally distributed. Data on HbA1c levels missing in 514 individuals. Data on smoking status at baseline missing in 20,173 individuals. BMI; body mass index, BP; blood pressure, GFR; glomerular filtration rate, LDL; low density lipoprotein.

**Table 6. Logistic regression model predicting the odds for developing carpal tunnel syndrome in people with diabetes type 2 (n = 84,184).**

| | Odds Ratio (95% CI) | P-value |
|---|---|---|
| *Model 1* | | |
| Age | 0.99 (0.99–0.99) | <0.001 |
| Male | *Reference* | |
| Female | 1.89 (1.75–2.04) | <0.001 |
| No retinopathy | *Reference* | |
| Retinopathy | 1.43 (1.27–1.62) | <0.001 |
| *Model 2* | | |
| Age | 0.98 (0.98–0.99) | <0.001 |
| Male | *Reference* | |
| Female | 1.84 (1.67–2.02) | <0.001 |
| No retinopathy | *Reference* | |
| Retinopathy | 0.93 (0.81–1.08) | 0.34 |
| HbA1c <48mmol/mol | *Reference* | |
| HbA1c 48–64 mmol/mol | 1.24 (1.10–1.39) | <0.001 |
| HbA1c >64 mmol/mol | 1.29 (1.11–1.49) | <0.001 |
| Non-smoker | *Reference* | |
| Smoker | 1.11 (0.98–1.25) | 0.11 |
| Diabetes duration | 1.06 (1.05–1.06) | <0.001 |
| Systolic blood pressure | 0.99 (0.99–0.99) | 0.02 |
| BMI | 1.03 (1.03–1.04) | <0.001 |
| LDL | 0.97 (0.91–1.03) | 0.31 |

BMI; body mass index, LDL; low density lipoprotein.

**Table 7. Logistic regression model predicting the odds for developing ulnar nerve entrapment at the elbow in people with diabetes type 2 (n = 84,184).**

|  | Odds Ratio (95% CI) | P-value |
|---|---|---|
| *Model 1* | | |
| **Age** | 0.98 (0.98–0.99) | **<0.001** |
| **Male** | *Reference* | |
| **Female** | 0.88 (0.75–1.04) | 0.14 |
| **No retinopathy** | *Reference* | |
| **Retinopathy** | 1.32 (1.01–1.71) | **0.04** |
| *Model 2* | | |
| **Age** | 0.99 (0.98–0.99) | **<0.001** |
| **Male** | *Reference* | |
| **Female** | 0.99 (0.81–1.23) | 0.99 |
| **No retinopathy** | *Reference* | |
| **Retinopathy** | 1.02 (0.74–1.40) | 0.90 |
| **HbA1c <48mmol/mol** | *Reference* | |
| **HbA1c 48–64 mmol/mol** | 1.14 (0.89–1.47) | 0.31 |
| **HbA1c >64 mmol/mol** | 1.58 (1.17–2.14) | **0.003** |
| **Non-smoker** | *Reference* | |
| **Smoker** | 1.72 (1.36–2.17) | **<0.001** |
| **Diabetes duration** | 1.03 (1.02–1.04) | **<0.001** |
| **Systolic blood pressure** | 0.99 (0.98–1.00) | **0.01** |
| **BMI** | 1.04 (1.02–1.06) | **<0.001** |
| **LDL** | 0.96 (0.84–1.10) | 0.54 |

BMI; body mass index, LDL; low density lipoprotein.

The exact mechanism for retinal degeneration in diabetic retinopathy and neural degeneration in peripheral neuropathy is still not fully understood. Retinopathy results from oxidative stress and inflammation as well as a deterioration of the blood retina barrier [19], and the pathophysiology of peripheral neuropathy may be similar, where diabetes leads to increased

**Table 8. Correlation matrix.** Diabetes-related factors and carpal tunnel syndrome and ulnar nerve entrapment at the elbow in people with diabetes type 2 (n = 84,184).

|  | CTS | UNE | Age | Retinopathy | HbA1c | Smoking | Duration of diabetes | Systolic blood pressure | BMI | LDL | Creatinine |
|---|---|---|---|---|---|---|---|---|---|---|---|
| **CTS** | 1 | | | | | | | | | | |
| **UNE** | 0.156 | 1 | | | | | | | | | |
| **Age** | -0.016 | -0.021 | 1 | | | | | | | | |
| **Retino-pathy** | 0.017 | 0.005 | 0.076 | 1 | | | | | | | |
| **HbA1c** | 0.04 | 0.024 | -0.036 | 0.119 | 1 | | | | | | |
| **Smoking** | 0.012 | 0.024 | -0.179 | -0.019 | 0.071 | 1 | | | | | |
| **Duration of diabetes** | 0.088 | 0.02 | 0.385 | 0.266 | 0.314 | -0.024 | 1 | | | | |
| **Systolic BP** | -0.01 | -0.006 | 0.189 | 0.034 | 0.041 | -0.03 | 0.075 | 1 | | | |
| **BMI** | 0.044 | 0.025 | -0.265 | -0.008 | 0.103 | -0.018 | -0.086 | 0.014 | 1 | | |
| **LDL** | -0.003 | -0.001 | -0.085 | -0.049 | 0.021 | 0.031 | -0.106 | 0.089 | 0.004 | 1 | |
| **Creati- nine** | -0.01 | -0.002 | 0.265 | 0.106 | 0.028 | -0.076 | 0.209 | 0.036 | -0.028 | -0.092 | 1 |

Pearson correlations. BMI; body mass index, BP; blood pressure, CTS; carpal tunnel syndrome, LDL; low density lipoprotein, UNE; ulnar nerve entrapment at the elbow.

inflammation in the neural tissue as well as microvascular damage to the vasa nervorum, leading in turn to ischemia of the nerve and eventually neuropathy [4]. Mean HbA1c levels were higher in the groups with retinopathy, than in the groups without. This reflects the fact that a higher level of HbA1c is associated with more complications. Higher levels of HbA1c cause an increase in oxidative stress and inflammation that has a negative effect on the endothelium of blood vessels in the retina, eventually resulting in retinopathy [20, 21]. Advanced retinopathy has also been associated with both neuropathy and nephropathy in patients with DM1 [18].

There are several other possible explanations for why hyperglycaemia increases the risk of developing EN [22]. High levels of blood glucose prompt the mitochondria in the nerve cell to produce reactive oxygen species that cause damage to the nerve [23]. This was demonstrated in one study from 2009, where the density of myelinated nerve fibres in patients with CTS was examined using light microscopy images. In patients with diabetes and CTS the density of myelinated nerve fibres in the posterior interosseous nerve was reduced in comparison to those with CTS and no current diagnosis of diabetes. The authors also found duplication of the basement membrane in the capillaries within the fascicle in the diabetic nerves, possibly resulting in a reduced blood flow to the nerves [24]. Compression itself also contributes to the pathology of EN in patients with diabetes. Oxidative reactions between glucose and collagen lead to an increase in AGEs, which cross-link collagen fibres in structures such as the transverse carpal ligament in the carpal tunnel [25]. The cross-linking results in a stiffening of the connective tissue. As hyperglycaemia actuates the sorbitol pathway, where sorbitol functions as an osmotic agent, fluid may be transported into the nerve cells, ultimately causing nerve trunk swelling. This, together with the impaired compliance of the ligament of the carpal tunnel as well as the cubital tunnel itself, may increase the build-up of pressure on the nerves causing EN [26]. Evidence of this is supplied by previous studies on the cross-section area (CSA) of the median and tibial nerves in patients with DM. The CSA in both nerves was significantly larger than the CSA in nerves of healthy controls, possibly as a result of intraneural oedema [27, 28].

Peripheral nerves are also negatively affected by higher levels of HbA1c [4]. In this study, a HbA1c of above 64mmol/mol significantly increased the likelihood of developing both CTS and UNE. This is in accordance with the results from the DCCT/EDIC trials which showed that a reduction of HbA1c reduces the prevalence of neuropathy in patients with DM1 [4].

We found that a higher BMI increases the risk of developing both CTS and UNE in both types of diabetes. A higher BMI is a known risk factor for both DM2 and cardiovascular disease, as a part of the metabolic syndrome MetS, defined by the International Diabetes Federation as central obesity in combination with elevated blood glucose, dyslipidaemia, and hypertension [4, 29, 30]. Abnormal levels of blood lipids elevate the concentration of LDL-molecules that infiltrate the arterial wall, causing inflammation there that gradually develops into atherosclerosis. As the thickness of the artery wall increases, the lumen decreases with a restricted blood flow as the consequence [31]. Nerves depend on high blood flow and are sensitive to ischemia. The disturbances of the microcirculation to the nerve, caused by the damaged arterial walls, may eventually contribute to the development of neuropathies [5]. Increased levels of LDL are a known risk factor for the development of CTS as high levels of LDL lead to increased oxidative stress which damages the nerves [16, 32]. In our logistic regressions however, LDL was not a significant predictor for the development of either CTS or UNE. Callaghan et al. [32] discussed the role of both cholesterol and free fatty acids in DM2, where cholesterol in particular has been shown to cause apoptosis in neurons, thus contributing to the development of neuropathies. It is worth mentioning that the levels of LDL in all the individuals in the present cohort were well regulated as the mean LDL in all sub-groups analysed was below or around 2.5mmol/l. According to Swedish guidelines, LDL in patients with

diabetes with no other cardiovascular risk factors should be kept below 2.5mmol/l [33]. It is possible that the results would be different in a population with more dysregulated blood lipid levels.

Furthermore, smoking, a cause of oxidative stress, is known to contribute to the inflammation and oxidation of LDL leading to a build-up of atherosclerosis and causing direct damage to the walls of the blood vessel [31, 34]. Smoking has also been shown to be an independent risk factor for the development of neuropathy in patients with DM1 [15]. In the present study, smoking was a significant risk factor for the development of UNE in DM2, possibly because of the negative effect that cigarette smoke has on vessels supplying blood to the nerves in the retina and peripheral nerves.

Previous research shows that a longer duration of DM is associated with more severe complications such as neuropathies, which correlates well with the result of this study [17]. A longer diabetes duration entails exposure to elevated blood glucose and other related risk factors over a longer period of time, allowing the pathological processes of retinopathy to develop. This is shown in the report from Pemp et al. [35], where longer duration of diabetes was associated with a reduction in total macular volume and inner neural layer volume.

## Strengths and limitations

This study has several strengths, the main one being the large study population. Previous comparable studies have had populations of well below 2,000 patients [4, 18, 35] whereas this study draws on data from over 80,000 individuals. The large study population enabled comparisons to be made between patients with DM1 and DM2. It seems that previous research on retinopathy in correlation with neuropathy, has primarily focused on patients with DM1, leading to a lack of literature about the occurrence of neuropathy in patients with DM2.

One limitation of our study is that the available data did not reveal the degree of severity of retinopathy. Other similar studies performed regarding correlations between retinopathy and neuropathies, have primarily focused on a specific degree of retinopathy such as non-progressive diabetic retinopathy. There is also a risk of selection bias as those who cannot manage their diabetes might not seek medical care and are therefore naturally not included in the NDR.

## Conclusion

In conclusion, retinopathy and CTS and UNE are correlated, but the correlation seems to be mediated through diabetes duration. Higher levels of HbA1c, longer diabetes duration, and high BMI are risk factors for developing both CTS and UNE and play a significant role in the development of EN when retinopathy is present.

## Acknowledgments

We are very grateful to the participating patients, units and staff working with the registries who made this research possible. We would also like to thank Tina Folker for her administrative support and Pat Shrimpton for the language revision.

## Author Contributions

**Conceptualization:** Malin Zimmerman.

**Data curation:** Caroline Olsson, Mattias Rydberg, Malin Zimmerman.

**Formal analysis:** Caroline Olsson, Mattias Rydberg, Malin Zimmerman.

**Funding acquisition:** Malin Zimmerman.

**Investigation:** Caroline Olsson.

**Methodology:** Malin Zimmerman.

**Project administration:** Malin Zimmerman.

**Resources:** Malin Zimmerman.

**Supervision:** Malin Zimmerman.

**Writing – original draft:** Caroline Olsson.

**Writing – review & editing:** Mattias Rydberg, Malin Zimmerman.

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
