## [Decision Letter · Decision Letter 0]

11 Jul 2022

PONE-D-22-17263Diabetic retinopathy as a predictor for peripheral compression neuropathies, a registry-based studyPLOS ONE

Dear Dr. Zimmerman,

Thank you for submitting your manuscript to PLOS ONE. After careful consideration, we feel that it has merit but does not fully meet PLOS ONE’s publication criteria as it currently stands. Therefore, we invite you to submit a revised version of the manuscript that addresses the points raised during the review process.

We look forward to receiving your revised manuscript.

Kind regards,

Alok Raghav, PhD

Academic Editor

PLOS ONE

Journal Requirements:

Reviewers' comments:

Reviewer's Responses to Questions

**Comments to the Author**

1. Is the manuscript technically sound, and do the data support the conclusions?

Reviewer #1: Yes

Reviewer #2: Yes

2. Has the statistical analysis been performed appropriately and rigorously? 

Reviewer #1: Yes

Reviewer #2: Yes

3. Have the authors made all data underlying the findings in their manuscript fully available?

Reviewer #1: Yes

Reviewer #2: Yes

4. Is the manuscript presented in an intelligible fashion and written in standard English?

Reviewer #1: Yes

Reviewer #2: Yes

5. Review Comments to the Author

Reviewer #1: The purpose of the study was to investigate if retinopathy can be used as a factor predicting the development of CTS and UNE. The Authors have used a large cohort based on which the results obtained are convicting. The advantage of having access to 95,437 individuals from the National Diabetes Registry is promising to conduct some additional analysis based on correlation analysis of variables that authors have. It will be worthwhile to do such analysis on the available dataset to make the study more interesting for the groups working on diabetes and its complication. Therefore, it is suggested to perform correlation analysis among the variables to identify the factors that might be contributing to diabetes induced CTS and UNE.

Reviewer #2: Authors have presented the study well. Authors have focused on developing predictors of peripheral compression neuropathy using diabetic retinopathy. The study is adding knowledge to the currently available domain and is also useful for the countries like developing nations where the prevalence of diabetic patients is increasing.

6. PLOS authors have the option to publish the peer review history of their article (what does this mean?). If published, this will include your full peer review and any attached files.

Reviewer #1: No

Reviewer #2: No

---

## [Author Response · Author response to Decision Letter 0]

7 Sep 2022

Dear Editor and Reviewers,

Thank you for your comments. We address them below. Please see changes in the revised manuscript. We have also reviewed the formatting requirements and the reference list. We hope that you will find the revised manuscript suitable for publication.

On behalf of all authors,

Malin Zimmerman, MD, PhD

Reviewer #1: The purpose of the study was to investigate if retinopathy can be used as a factor predicting the development of CTS and UNE. The Authors have used a large cohort based on which the results obtained are convicting. The advantage of having access to 95,437 individuals from the National Diabetes Registry is promising to conduct some additional analysis based on correlation analysis of variables that authors have. It will be worthwhile to do such analysis on the available dataset to make the study more interesting for the groups working on diabetes and its complication. Therefore, it is suggested to perform correlation analysis among the variables to identify the factors that might be contributing to diabetes induced CTS and UNE.

Reply: Thank you for the suggestion. We have added a correlation analysis, please see table 4 and table 8. 

Reviewer #2: Authors have presented the study well. Authors have focused on developing predictors of peripheral compression neuropathy using diabetic retinopathy. The study is adding knowledge to the currently available domain and is also useful for the countries like developing nations where the prevalence of diabetic patients is increasing.

Reply: Thank you! We hope that our study will help in the understanding of diabetes complications.

---

## [Editor Report · Decision Letter 1]

20 Sep 2022

Diabetic retinopathy as a predictor for peripheral compression neuropathies, a registry-based study

PONE-D-22-17263R1

Dear Dr. Zimmerman,

We’re pleased to inform you that your manuscript has been judged scientifically suitable for publication and will be formally accepted for publication once it meets all outstanding technical requirements.

Kind regards,

Alok Raghav, PhD

Academic Editor

PLOS ONE
---

## [Editor Report · Acceptance letter]

22 Sep 2022

PONE-D-22-17263R1 

Diabetic retinopathy as a predictor for peripheral compression neuropathies, a registry-based study 

Dear Dr. Zimmerman:

I'm pleased to inform you that your manuscript has been deemed suitable for publication in PLOS ONE. Congratulations! Your manuscript is now with our production department. 

Kind regards, 

on behalf of

Dr. Alok Raghav 

Academic Editor

PLOS ONE